# Compact High-$T_c$ Superconducting Terahertz Emitter with Tunable Frequency from 0.15 to 1 THz

Hancong Sun [1,2], Shixian Chen [2], Yong-Lei Wang [1,2], Guozhu Sun [1,2], Jian Chen [1,2], Takeshi Hatano [3], Valery P. Koshelets [4], Dieter Koelle [5], Reinhold Kleiner [5], Huabing Wang [1,2,*] and Peiheng Wu [1,2]

1 Purple Mountain Laboratories, Nanjing 211111, China
2 Research Institute of Superconductor Electronics, Nanjing University, Nanjing 210023, China
3 National Institute for Materials Science, Tsukuba 3050047, Japan
4 Kotel'nikov Institute of Radio Engineering and Electronics, Moscow 125009, Russia
5 Physikalisches Institut, Center for Quantum Science (CQ) and LISA+, Universität Tübingen, 72076 Tübingen, Germany
* Correspondence: hbwang@nju.edu.cn

**Abstract:** A compact cryogenic terahertz emitter is highly desired for applications in terahertz astronomy with a broad frequency range of emissions and relatively high emission power. In this paper, we report on a terahertz emitter based on $Bi_2Sr_2CaCu_2O_{8+\delta}$ (BSCCO) intrinsic Josephson junctions, with a frequency range from 0.15 to 1.01 THz. The emitter is a square gold-BSCCO-gold mesa on a sapphire substrate fabricated by a simple and efficient method. The highest emission power of 5.62 μW at 0.35 THz was observed at 50 K. A record low emission frequency of 0.15 THz was achieved at 85 K, extending the applicability of BSCCO terahertz emitters in the low-frequency range.

**Keywords:** BSCCO; superconducting terahertz emitter; low-frequency end of terahertz emitter; intrinsic Josephson junctions

## 1. Introduction

The terahertz (THz) technology has been investigated intensively for decades due to its great potential in fundamental and applied research fields, such as high-bandwidth communication, environmental monitoring and public security [1,2]. Compact intrinsic-Josephson junction (IJJ) THz sources made of the high-$T_c$ superconductor $Bi_2Sr_2CaCu_2O_{8+\delta}$ (BSCCO) have attracted much attention for their continuous and tunable terahertz radiation at the sub-THz range [3,4].

$Bi_2Sr_2CaCu_2O_{8+\delta}$ naturally forms serial stacks of IJJs, with about 670 Josephson junctions per micron of the stack thickness [5]. For IJJ-THz emitters, radiation is generated by the ac Josephson effect. The supercurrent across the stack biased at a voltage $V$ can oscillate synchronously at a frequency $f = V/N\Phi_0$, where $\Phi_0$ is the flux quantum, and $N$ is the number of junctions in the stack. For a single junction, the emission power is typically on the order of pW to nW. To reach a high emission power, $N$ should be large, and the IJJs in the stack should be synchronized. In 2007, coherent terahertz emission was first detected in a 1 μm-thick BSSCO stack, with an emission power of up to 0.5 μW and emission frequencies between 0.5 and 0.85 THz [6]. Since then, IJJ-THz emitters have been studied intensively, both experimentally and theoretically [6–31]. Single stacks with lateral dimensions on the order of a few 100 μm and consisting of around 1000 IJJs now reach an emission frequency of up to 2.4 THz [29] and an emission power of about tens of microwatts [13,22]. The maximum power of about 610 μW was detected from a three-stack array [16]. Demonstrations of applications in terahertz imaging and detection have been given [10,15,20,22,26].

Theoretically, the frequencies of emission from a stack can be tuned by the applied voltage from a small value to more than 20 THz [3], limited by the gap energy of the IJJs.

This, in principle, makes BSCCO emitters to be ideal terahertz sources for the application of wide-band continuous waves. However, the realized frequency range of BSCCO emitters is far from the theoretical values. Much attention has been paid to extending the emission frequency range of BSCCO emitter towards high frequencies. The maximum emission frequency of typical BSCCO emitters with more than 600 IJJs and lateral sizes over 100 μm was always limited by self-heating, which leads to the formation of a hot spot in the stack [12], decreasing the voltage across the Josephson junction in the stack. In order to improve the emission frequency, sandwich-structure devices were created for good cooling [24,25,29,30], with the highest emission frequency being improved to values above 2 THz [24,29]. Recently, emission frequencies between 1 and 11 THz have been observed by fabricating smaller stacks and using an on-chip detector [32].

On the low-frequency side of emission, the Josephson plasma frequency ($f_{pl}$) and resonant cavity modes play important roles in limiting the emission frequency. The $f_{pl}$ depends on temperature $f_{pl}(T) \propto \sqrt{j_c(T)}$ and determines a cutoff emission frequency of about $4 f_{pl}$ [11]. At low bath temperatures $T_b$, $f_{pl}$ is saturated, leading to a constant cutoff emission frequency at the low-frequency side. With an increase in $T_b$, $f_{pl}$ decreases and approaches zero when $T_b$ is close to $T_c$, leading to a reduced low-frequency cutoff [11,33]. Further, the excitation of collective cavity resonances of the whole stack has been shown to enhance terahertz emission [6–9,11,17,21,23–25,29,34], with the emission frequency depending on the resonance mode. Therefore, BSCCO stacks exhibiting low-frequency resonance modes are likely to emit terahertz waves at reasonable emission power at the low-frequency end of the spectrum. However, unlike the high-frequency emission properties, emission at the low-frequency end has received less attention so far. The reported lowest values of emission frequency were always above 0.3 THz [9,21,23–25,29,30], which limits the applicability of the BSCCO emitters on the low-frequency side.

To extend the emission frequency range to lower frequencies, we fabricated square IJJ stacks with a lateral size of 240 × 240 μm². Two stacks of these sizes were measured, and here, we report data from one of them. The other stack behaved similarly. The current–voltage and emission properties were studied from 5 to 85 K. Emission was observed up to 1.01 THz for $T_b$ = 5 K. A record low emission-frequency band of 0.15 to 0.3 THz was observed at bath temperatures between 75 and 85 K.

## 2. Sample Preparation and Measurement Techniques

High-quality BSCCO single crystals with a transition temperature $T_c$ of 88.6 K were grown by a floating-zone technique. The main steps of the fabrication process are shown in Figure 1a–d. The process is similar to that reported in Ref. [24]. Here, we describe this process in more detail. First, a piece of a BSCCO single crystal was fixed on a weakly adhesive double-sided tape on the holder of an evaporator. Immediately after cleaving the crystal with another tape, a 120 nm-thick gold film was deposited on its upper surface, as shown in Figure 1a. Next, the crystal with the gold layer was removed from the double-sided tape by a thin wooden sheet and turned over. The sample was then glued onto a sapphire substrate by a thin layer of polyimide. After two hours of drying at 110 °C, the excess glue on the edge of the sample was removed to allow a further cleaving step by tape. Immediately after cleaving, the fresh surface was covered with a 120 nm-thick gold layer, forming the GBG structure (see Figure 1b). Using photolithography, a 240 × 240 μm² square pattern on the mask was transferred to the gold layer, and then the sample was etched down to the lower gold layer by ion milling, as shown in Figure 1c. To pattern the bottom electrode, we manually used a whittled toothpick to paste the photoresist onto the bottom gold film and the stack and then removed the unprotected gold layer by ion milling. After the ion milling process, the photoresist was removed in an acetone solution. The final BSCCO stack on the sapphire substrate is shown in Figure 1d, with lateral dimensions of 240 × 240 μm² and a thickness of about 1.8 μm, corresponding to about 1200 IJJs in the stack. Finally, polyimide was pasted manually to protect the side walls of the stack from shorts, and silver epoxy was used to connect two gold wires to the electrodes.

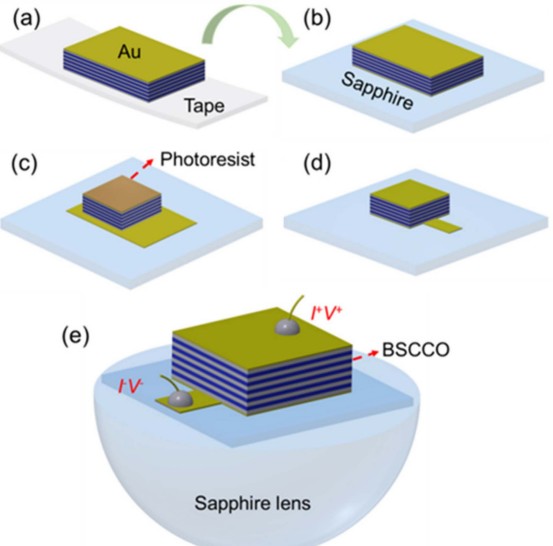

**Figure 1.** (**a–d**) Schematic diagram of the main steps in the fabrication process. (**e**) Schematic view of the final BSCCO stack mounted on a sapphire lens.

After microfabrication, the sample was glued onto a hemispherical sapphire lens with a diameter of 6 mm and placed into a helium flow cryostat with a controlled temperature from 4.2 K to room temperature. The terahertz emission from the BSSCO stack was detected with a TYDEX GC-1P Golay cell, and the emission frequency was measured by a terahertz interferometer [35]. The frequency resolution of the interferometer was about 7.5 GHz, which is calculated by $\Delta f = c/(2d_{max}\cos\theta)$, where $d_{max} = 20$ mm is the maximum differential displacement of the lamellar mirror, $\theta \approx 4.6°$ is the angle of incidence at the lamellar mirror and $c$ is the speed of light. Here, we should also note that the device reported here is the pristine stack, and sample 1 reported in Ref. [11] is a modified device, where all the side walls were protected by the polyimide, and a 100 nm gold film was evaporated on the top of the stack for reflecting the terahertz radiation into the lens.

## 3. Results

Figure 2a shows the current–voltage curves (IVCs) at $T_b$ = 5 K, 50 K and 70 K. The contact resistance was subtracted. The gray arrows in Figure 2a indicate the sweep direction of the bias current across the stack at $T_b$ = 5 K. For this bath temperature, all junctions switched to their resistive state when the bias current was increased from zero to 60.5 mA. By further increasing the bias current to 72.6 mA and then decreasing it to zero, the S-shaped IVC was obtained. The maximum voltage of the emitter is about 2.88 V, corresponding to about 2.4 mV across each Josephson junction, indicating that the heat exhaust of this device was efficient. As the bath temperature was increased, the back-bending of the IVCs was reduced. It was absent for $T_b$ = 70 K. The shapes of IVCs at different bath temperatures are similar to those in Ref. [24], where we also performed 3D numerical simulations of the stack dynamics to better understand the observed IVCs.

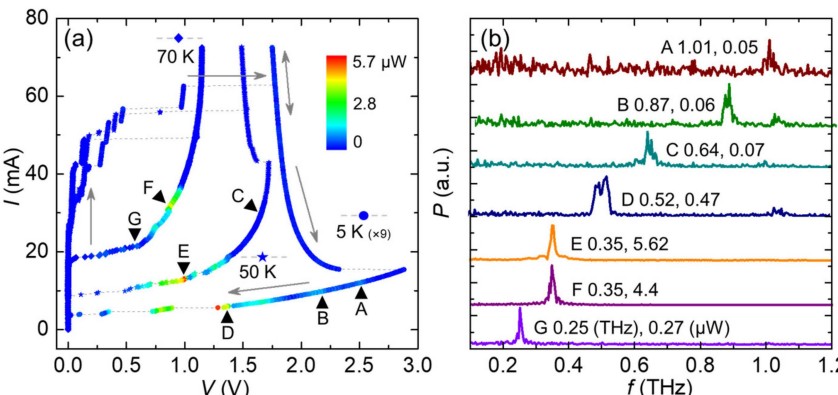

**Figure 2.** (**a**) Transport and terahertz emission characteristics of the BSCCO stack at $T_b$ = 5 K, 50 K and 70 K. The gray arrows indicate the sweep direction of bias current for the 5-K curve. Colors indicate the emission power (multiplied by a factor of 9 for the $T_b$ = 5 K data) detected by the Golay cell. (**b**) Fourier spectra of the emission for the bias points *A–G* indicated in (**a**); the numbers next to the labels are the values of emission peak frequencies and emission power. The data in (**b**) are rescaled to yield similar heights of the emission peaks.

The emission power and frequency were also characterized at $T_b$ = 5 K, 50 K and 70 K, as shown in Figure 2. The maximum emission power of about 5.62 µW was detected at point *E* (*I* = 12.96 mA, *V* = 0.992 V) at $T_b$ = 50 K, with a measured emission frequency of 0.35 THz. For this bath temperature, the highest emission frequency was observed to be 0.64 THz, measured at current *I* = 30.03 mA (point *C* in Figure 2a,b). At $T_b$ = 70 K, the highest emission power of 4.4 µW was found at bias point *F* (*I* = 31.87 mA, *V* = 0.877 V), with an emission frequency of 0.35 THz, corresponding to about 1230 junctions in the stack. This is nearly the same junction number as the value we estimated from the crystal thickness. Furthermore, within experimental accuracy, the junction number did not change with the bath temperature. The lowest emission frequency was measured to be 0.25 THz at *I* = 21.17 mA (point *G* in Figure 2a,b), indicating the good low-frequency performance of the stack. For $T_b$ = 5 K, the emission occurs mainly in the low-bias region, with a wide voltage range from 2.56 to 0.34 V. For the stable working points, the highest emission power of about 0.47 µW occurs at point *D* (*I* = 5.96 mA, *V* = 1.36 V), with an emission frequency of 0.52 THz. At bias point *B* (*I* = 9.92 mA, *V* = 2.2 V), the emission frequency is measured to be 0.87 THz. The highest emission frequency was observed to be 1.01 THz at *I* = 12.28 mA and *V* = 2.541 V (point *A* in Figure 2a,b), indicating the good cooling of the emitter.

In Figure 3, we studied the IVCs and emission characteristics at $T_b$ = 75 K, 80 K and 85 K. For the three bath temperatures, the IVCs exhibit a positive differential resistance at their resistance state. For $T_b$ = 75 K, the maximum emission power of about 3.5 µW was observed at *I* = 24.79 mA and *V* = 0.578 V. At point *C* (*I* = 25.46 mA and *V* = 0.609 V), the measured emission power was about 2.6 µW, with an emission frequency of 0.23 THz. Spectra for the bias points *A* and *B* in Figure 3a are also measured, as shown in Figure 3b. The emission frequencies are 0.33 THz and 0.31 THz for points *A* and *B*, respectively. At $T_b$ = 80 K, a maximum emission power of about 4.6 µW was obtained at *I* = 28.46 mA and *V* = 0.541 V. Low frequencies of 0.23 THz and 0.2 THz were obtained at *I* = 29.82 mA and 26.13 mA (point *D* and *E*), respectively. The lowest emission frequency, *f* = 0.15 THz, with an emission power of about 0.18 µW, was found at *I* = 23.39 mA (point *F* in Figure 3a,b at $T_b$ = 85 K. To our knowledge, 0.15 THz is a record for the low-frequency end of emission of BSCCO emitters.

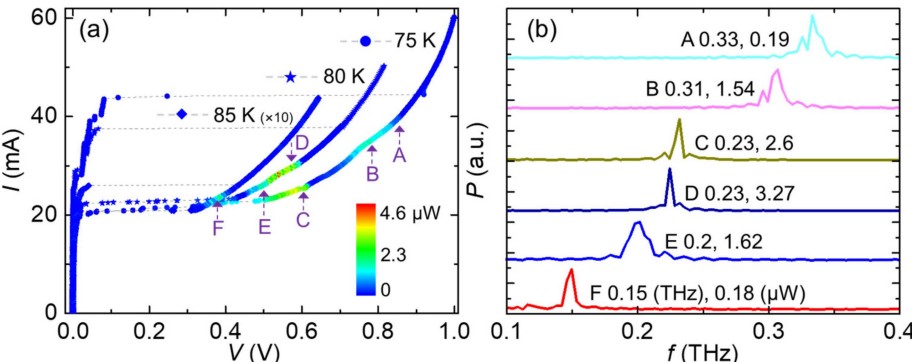

**Figure 3.** (**a**) Transport and terahertz emission characteristics of the BSCCO stack at $T_b$ = 75 K, 80 K and 85 K. Colors indicate the emission power (multiplied by a factor of 10 for the $T_b$ = 85 K data) detected by the Golay cell. (**b**) Fourier spectra of the emission for the bias points *A–F* indicated in (**a**); the numbers next to the labels are the values of emission peak frequencies and emission power. The data in (**b**) are rescaled to yield similar heights of the emission peaks.

Figure 4 displays the radiation properties of the BSCCO emitter at $T_b$ from 5 to 85 K. Over the whole range of operating temperatures, the emission frequencies are between 0.15 and 1.01 THz. A high emission power is obtained mainly in the frequency range between 0.2 and 0.52 THz, as shown in Figure 4a, indicating the desired good low-frequency performance of the emitter. Four obvious emission peaks near 0.21 THz, 0.32 THz, 0.42 THz and 0.5 THz appear on the radiation intensity spectrum, as indicated by the dashed lines in Figure 4a. At these peaks, it is likely that cavity resonances are excited. For a square IJJ stack, the resonances occur at frequencies $f_r = (c_0/2n_rL)\sqrt{k^2 + l^2}$, where $k$ and $l$ are integers, $L$ = 240 µm is the length of the square stack, $c_0$ is the velocity of light in a vacuum, and $n_r \approx 4.2$ is the refractive index of BSCCO crystals [7,8]. According to the above equation, the cavity frequency of TM (1, 1) resonance modes is about 0.21 THz, which has a good match with the emission peak near 0.21 THz in Figure 4a. The highest emission peak with measured frequencies of about 0.32 THz, as shown in Figure 4a, could match the predicted TM (2, 0) or TM (0, 2) resonance mode with a cavity frequency of about 0.3 THz. The predicted TM (2, 2) resonance modes with a cavity frequency of about 0.42 THz also match well with the peak near 0.42 THz. For the peak near 0.5 THz, the TM (1, 3) or TM (3, 1) resonance mode with the predicted cavity frequency of about 0.47 THz may have been excited to enhance the terahertz radiation. We note that, according to Ref. [34], degenerate cavity modes should not lead to an enhancement of emission, at least for homogeneous stacks. However, we operate our device at high bath temperatures. Here, even small gradients in the stack temperature could lead to gradients in the Josephson critical current density that are large enough to break this symmetry.

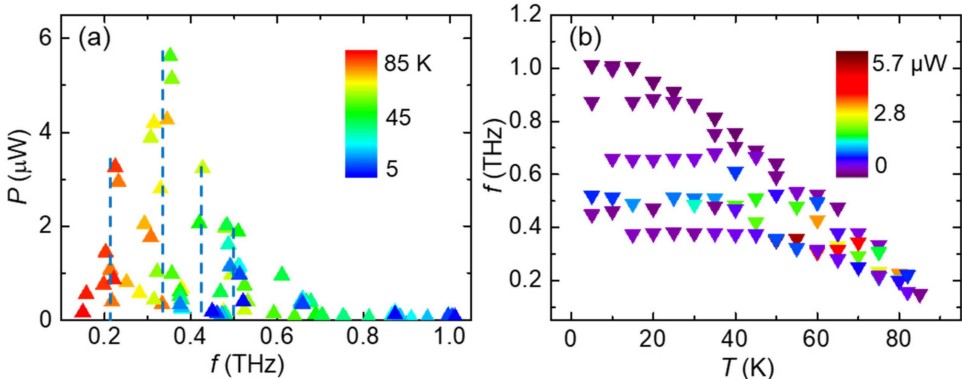

**Figure 4.** (**a**) The detected radiation power versus the measured frequency at the color-coded $T_b$ values. Four radiation peaks are marked by dashed lines. (**b**) The measured radiation frequency versus $T_b$ from 5 to 85 K. The radiation power is indicated by the colors.

Figure 4b shows the observed emission $f$ versus $T_b$ between 5 and 85 K, using the emission power as color code. At low bath temperatures, the emitter exhibits a wide frequency tunability. At $T_b$ = 15 K, the emission $f$ was observed, ranging from 0.374 to 1.004 THz. A clear cutoff emission frequency of about 0.37 THz in Figure 4b can be found for $T_b \leq 45$ K, which is mainly caused by the saturated $f_{pl}$. As $T_b$ is increased, the measured lowest emission frequency of each bath temperature decreases due to the decreasing $f_{pl}$. For $T_b \geq 80$ K, the measured emission frequencies are lower than 0.25 THz, with the lowest emission frequency being 0.158 THz and 0.15 THz at $T_b$ = 82 K and 85 K, respectively.

## 4. Conclusions

In summary, we designed and fabricated a square IJJ stack consisting of about 1200 junctions with a lateral size of 240×240 μm². The goal was to improve the emission properties of the BSCCO emitters on the low-frequency end of emission, making use of low-lying geometric resonances of the stack. We studied the current–voltage characteristics and achieved continuous terahertz emission with frequency from 0.15 to 1.01 THz at bath temperatures ranging from 5 to 85 K. We observed the most powerful emission in the frequency range between 0.2 and 0.52 THz. A record-low emission frequency of 0.15 THz was observed at $T_b$ = 85 K. The geometry studied can be used to provide the applicability for BSCCO terahertz emitters also at frequencies well below 0.4 THz.

**Author Contributions:** Conceptualization, H.S. and H.W.; methodology, H.S., S.C. and H.W.; validation, Y.-L.W., G.S., J.C. and H.W.; formal analysis, H.S., S.C. and H.W.; investigation, H.S., S.C. and H.W; resources, T.H., Y.-L.W., G.S., J.C. and H.W.; data curation, H.S. and S.C.; writing—original draft preparation, H.S. and S.C.; writing—review and editing, H.S., H.W., P.W., Y.-L.W., G.S., J.C., T.H., V.P.K., D.K. and R.K.; visualization, H.S. and S.C.; supervision, H.W. and P.W.; project administration, H.W. and P.W.; funding acquisition, H.W. and P.W. All authors have read and agreed to the published version of the manuscript.

**Funding:** This research was funded by the National Natural Science Foundation of China (Grant Nos. 61727805, 11961141002, 62288101, 62201396); the National Key R&D Program of China (Grant Nos. 2021YFA0718802, 2018YFA0209002); Jiangsu Key Laboratory of Advanced Techniques for Manipulating Electromagnetic Waves; the COST Action CA21144 superqumap; the USU "Cryointegral" that is supported by a grant from the Ministry of Science and Higher Education of the Russian Federation, agreement No. 075-15-2021-667; and the Innovation and Entrepreneurship Program of Jiangsu Province (Grant No. JSSCBS20211334).

**Institutional Review Board Statement:** Not applicable.

**Informed Consent Statement:** Not applicable.

**Data Availability Statement:** Not applicable.

**Acknowledgments:** We thank Zihan Wei, Dingding Li, Hongmei Du, Ping Zhang, Yangyang Lyu, Mei Yu, Jiazheng Pan and Xuecou Tu for the valuable discussions.

**Conflicts of Interest:** The authors declare no conflict of interest. H.C.S. and S.X.C. contributed equally to this work.

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
