# Peer review of "Compact High-Tc Superconducting Terahertz Emitter with Tunable Frequency from 0.15 to 1 THz"

_applsci, doi:10.3390/app13063469_

Round 1
Reviewer 2 Report
The authors report on a terahertz emitter intrinsic Josephson-junctions. A record low emission frequency of 0.15 THz was achieved at 85 K, extending the applicability of BSCCO terahertz emitters in the low-frequency range. The paper is sound and original, although simple and written for experts from the very narrow field and as such less interesting to the general audience of the journal. Since the results are novel and important, I recommend to publish the manuscript.
Author Response
We thank the reviewer for the positive comments.
Reviewer 3 Report
In the given paper a terahertz emitter based on a square BSCCO mesa-structure was designed and fabricated. The emission properties on the low-frequency end of emission was improved leading to record-low emission frequency of 0.15 THz at 85 K. Continuous terahertz emission with frequency from 0.15 to 1.01 THz at bath temperatures ranging from 5 to 85 K was achieved. The highest emission power of 5.62 μW at 0.352 THz was observed at 50 K.
I rate this work as a well-organized, high-tech and science-intensive work, which, in my opinion, undoubtedly deserves publication after discussing several questions and suggestions:
1. Although Ref. 35 provides data on the frequency resolution and general characteristics of the interferometer, I would recommend duplicating them in the text of the article, since this is very important for evaluating the results of the current paper.
2. Figure 2a is very difficult to read. It is impossible to distinguish a forward branch of IVC for different temperatures, although this is not very important in this work. I propose to highlight different temperatures with different symbols. In Figure 3, the symbols for the two temperatures are different (squares and circles). Thus, for each temperature, different characters can be selected: circles, stars, diamonds, etc.
3. For clarity, in Figures 2b and 3b it would be useful to label not only the center frequency of the radiation, but also the measured power at a given point.
4. The point with the number of junctions in the stack is not entirely clear. In chapter 2 it is written that the mesa thickness of about 1.8 μm corresponds to about 1200 IJJs in the stack. While in chapter 3 another number of 1230 IJJs is given for 70 K. The impression was that the number of junctions differs for different temperatures. Could you clarify this point? How was this number determined and what did it depend on?
5. Figure 4 is also difficult to read. It would be better to make the gradient more contrast and clear. For example, from blue color for low temperature to red color for 85 K, etc.
